# Applying a Sliding Mode Controller to Maximum Power Point Tracking in a Quasi Z-Source Inverter Based on the Power Curve of a Photovoltaic Cell

**Jaber Merrikhi Ahangarkolaei** [1] , **Mahdi Izadi** [2,*] **and Tohid Nouri** [1]

1   Department of Electrical Engineering, Sari Branch, Islamic Azad University, Sari 4815733971, Iran; jaber.merrikhi@gmail.com (J.M.A.); thdnouri@iausari.ac.ir (T.N.)
2   Department of Electrical Engineering, Firoozkooh Branch, Islamic Azad University, Firoozkooh 3981838381, Iran
*   Correspondence: aryaphase@yahoo.com

**Abstract:** Due to the nonlinear nature of photovoltaic (PV) cells and the dependence of the maximum achievable power on environmental conditions, a robust nonlinear controller is essential to warrant maximum power point tracking (MPPT) by managing the nonlinearities of the system and making it robust against varying environmental conditions. Most methods used to obtain MPPT have some disadvantages; one of them is the oscillation around the operating point. In this paper, to minimize these problems, a robust nonlinear sliding mode controller based on the power curve of a PV (SMC-PCPV) was proposed to determine the maximum power point (MPP) of a PV panel, for a quasi Z-source inverter (qZSI) as a single-stage inverter. Single-stage inverters have lower components and prices, smaller sizes, more simplicity, and higher efficiency than two-stage inverters. One of the important features of this controller is its ease of implementation compared to other methods presented in the articles. To show the effectiveness and robustness of the proposed scheme, the SMC-PCPV was carried out on computer simulations and laboratory prototypes. The simulation and experimental results showed that the proposed controller was properly resistant to changes in input parameters, such as temperature and radiation, and controlled the converter at the best point to obtain the most power from the PV panel, and it had good speed in response to the changing environmental condition.

**Keywords:** maximum power point tracking (MPPT); sliding mode controller (SMC); quasi Z-source inverter (qZSI); photovoltaics (PV)

## 1. Introduction

Currently, due to the fact of environmental issues and rising costs of fossil fuels, replacing traditional energy with renewable energy is a comprehensible necessity for everyone [1]. One renewable energy source that has been considered by researchers is solar energy. Among the various types of solar energy, photovoltaics (PV) has been considered because of its practical applications. However, the output power of solar cells is extremely dependent on environmental conditions such as temperature and radiation intensity [2,3].

Converter and inverters are the major devices for a PV power system, where traditional converters are typically formed by two power stages connected through a dc-link capacitor [4]. A dc/dc converter, as the first stage, is used to generate the dc bus voltage required by the inverter. The main role in absorbing maximum power from the PV panel is the responsibility of this dc/dc converter. To implement the dc/dc converter, various converters are used, which can be referred to as a flyback, push–pull, and resonant converters with two inductors and one capacitor (LLC). Moreover, the second stage is a dc/ac inverter that injects an ac current proportional to the power produced by the PV module [4–6].

Single-stage inverters have received more attention than two-stage inverters due to the fact of their fewer components and lower price, smaller size, simplicity, and higher efficiency [7–9]. The Z-source inverter (ZSI), as a single-stage power converter, has been reported with a step-up/down function in PV systems [10,11]. The discontinuous input current in ZSI in PV applications makes it impossible to meet the maximum energy harvesting; nevertheless, the quasi-Z-Source inverter (qZSI) has the same properties as ZSI, except that the input current is continuous [12,13]. In particular, the use of qZSI in PV system applications improves their performance and efficiency.

Due to the nonlinear characteristics of the voltage and current of a PV module, direct connection of the loads to PV module terminals probably leads to low PV power extraction and a nonoptimal operating point. Therefore, a controller, named a maximum power point tracker (MPPT), is needed to warrant that the maximum available power is obtained from PV cells. There are many different approaches to the MPPT algorithm, ranging from simple to advanced ones that have been reported in the literature in the ZSI inverter family [14,15]. The MPPT methods can be divided into two classes: those based on gradient classic methods and those based on intelligent methods. The most favorite classic algorithms for MPPT in PV systems are perturbation and observation (P&O), incremental conductance (IncCond), linearization around the MPP (LAMPP), and constant voltage (CV) [7,15–20]. Easy implementation and good performance in uniform weather conditions are the advantages of these methods. Among the MPPT approaches applied, P&O is one of the most common techniques, with ease its implementation and relatively good performance [17,18]. The defect of this method is that the operation point oscillates around the maximum power point (MPP), at a steady-state, which is proportional to the size of perturbation [19]. The IncCond method is one of the MPPT methods applying a step size control signal to discover the MPP. The step size of the decrement or increment determines the speed of reaching the MPP. By applying a larger step size, fast-tracking can be attained, but this can lead to fluctuations around the MPP. Applying the IncCond method is a trade-off between convergence speed and result accuracy [13,18]. In [7], to achieve the maximum power used, the small-signal model of the PV was used, which was obtained from the linearization of the $i_{pv} - v_{pv}$ characteristics around the MPP; however, its accuracy cannot be guaranteed if the linear model of the PV is operated at points other than the MPP. However, the P&O, IncCond, and LAMPP methods did not have good operation during the rapid changing of weather conditions [19].

Due to the problems mentioned above, researchers have worked on employing intelligence technologies [21]. In [8], the maximum power was achieved by an adaptive neuro-fuzzy inference system (ANFIS) based on MPPT for qZSI in standalone operation. In [22], the authors developed an enhancing ANFIS (EANFIS) algorithm composed of particle swarm optimization (PSO) for MPPT control in a qZSI to feed a brushless motor (BLCD). In [15], a BLDC motor was powered by PV through a ZSI. To achieve the MPPT of the PV, a fuzzy logic-incremental conductance (FL-IC) MPPT scheme was proposed. A model predictive control MPPT (MPC-MPPT) was developed in [23] for a qZSI-based grid-connected PV power system. However, these methods need big data for training as well as more memory space. Thus, the complexity of the implementation and the tendency to move toward the local MPP in the case of partial shading are the disadvantages [24,25].

Both qZSI and PV cells have nonlinear dynamics in nature; as a result, nonlinear control will perform better than a linear control against perturbations and changes in the parameter. The SMC is a type of nonlinear controller which was introduced to control systems having uncertainties. Its main superiority is guaranteed robustness and stability against parameter, input, and load uncertainties [26]. In addition, the implementation of the sliding mode (SM) controller is relatively easy compared to other kinds of nonlinear controllers, for the reason that being a controller, it has a high degree of flexibility in its design choices. The implementation of the SM controller to track the MPP has been reported in many articles using DC-DC converters [19,24,27,28]. Furthermore, the SM controller is widely applied to control grid-tie or off-grid qZSI [29–33]. In [29], an integral SM controller-

based strategy was proposed to control the battery charging current for unbalanced power compensation in qZSI with battery. A lack of mesh [32] was applied to the SM control to regulate qZSI capacitor voltage. Researchers in [31] offered a multi-input multi-output (MIMO) SM control for a qZSI.

The main aim of this paper was to absorb the maximum power of PV panels through an SM controller based on the power curve of a PV (SMC-PCPV) with a qZSI. In this article, effective integration of a PV panel into the qZSI is provided with the aid of the proposed controller, resulting in a faster response in the variable atmospheric conditions and minification of the oscillation around the operating point. The overall system included a PV panel connected to a qZSI that injected power into a load. The SM controller was derived based on the nonlinear mathematical model of a PV panel with a qZSI. The robustness of the SMC-PCPV was investigated in the presence of environmental changes, whereas the SMC-PCPV was implemented in MATLAB/Simulink and a laboratory prototype. A laboratory prototype based on a TMS320F28379D digital signal processor was constructed. The overall structure of the article is as follows: in Section 2, the PV characteristics are described; Section 3 develops a comprehensive model for single-phase qZSI; the SMC-PCPV is explained in Section 4; simulation and experimental results are presented in Section 5; finally, Section 6 summarizes the conclusions and future work.

## 2. PV Characteristics

The solar cell is a fundamental component of a PV system. Solar cells render the non-linear characteristics of P-V and I-V, which depend on the temperature of the cell and irradiance level ($G$). $G$ is usually referred to as the rated irradiation (i.e., $G$ = 1 Sun = 1000 W/m$^2$). The electrical equivalent circuit of a PV with a single diode was adopted. As shown in Figure 1, the equivalent circuit of a PV cell consists of a light-generated source, a diode connected in parallel, series resistances, and parallel resistances. The mathematical expression for the equivalent model that reflects the relationship between the current and voltage in the PV module can be written as:

$$i_{pv} = I_{ph} - I_d \left[ exp\left( \frac{q}{k_b TA} V_{pv} \right) - 1 \right] \tag{1}$$

$$I_{ph} = S[I_{scr} + k_i(T - T_r)] \tag{2}$$

$$I_d = I_{rr} \left[ \frac{T}{T_r} \right]^3 exp\left( \frac{qE_g}{kQA} \left[ \frac{1}{T_r} - \frac{1}{T} \right] \right) \tag{3}$$

where $i_{pv}$ and $v_{pv}$ are the output current and voltage ($A$, $V$); $T$ is the cell temperature ($K$); $S$ is the solar irradiance (W/m$^2$); $I_{ph}$ is the light-generated current; $I_{rr}$ is the saturation current at $T_r$; $E_g$ is the band-gap energy of the material; $T_r$ is the reference temperature; $q$ is the charge of an electron; Ki is the short circuit of the temperature coefficient; $I_d$ is the PV saturation current; $I_{scr}$ is the short circuit of the current at the reference condition; $k_b$ is the Boltzmann's constant; $Q$. is the total electron charge: $A$ is the ideality factor.

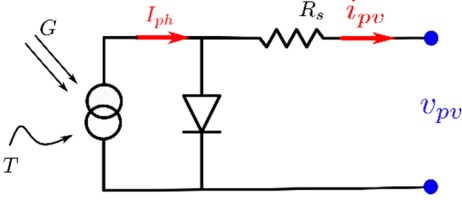

**Figure 1.** Equivalent circuit model of a PV.

Figure 2 shows the $i_{pv} - v_{pv}$ and $P_{pv} - v_{pv}$ characteristics of the Trina Solar TPS105S-85W-MONO, a typical 85 W PV panel under different irradiance levels and at a fixed temperature (25 °C). Figure 3 shows the influence of temperature on the MPP of the Trina Solar TPS105S-85W-MONO at $G$ = 1 Sun.

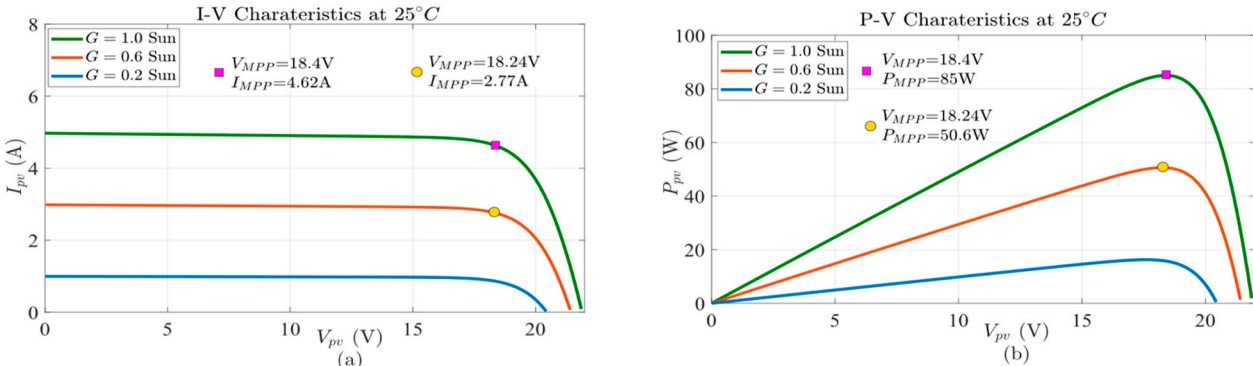

**Figure 2.** The effect of irradiance changes on the (**a**) I-V and (**b**) P-V characteristics.

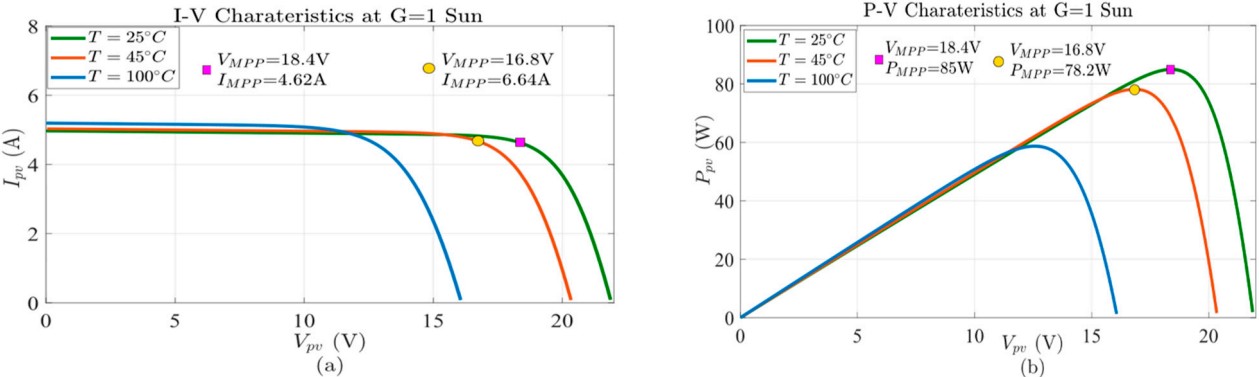

**Figure 3.** The effect of temperature change on the (**a**) I-V and (**b**) P-V characteristics.

Figures 2 and 3 show that the obtained power of PV is a nonlinear function of temperature, irradiance, and the current drawn from the panel. Thus, it is important to determine the point where the maximum power is received from the PV according to the environmental conditions.

## 3. Mathematical Model of the qZSI

A single-phase qZSI, as shown in Figure 4, is made up of a quasi-Z-source impedance network and a single-phase inverter, which operates as a single-stage converter [4]. The qZSI also has two modes of operation (i.e., the shoot-through state (ST) and the non-shoot-through state (NST)), which are demonstrated through two equivalent circuits in Figure 5. During the NST state, as shown in Figure 5a, the inverter acts as a traditional voltage source inverter [18].

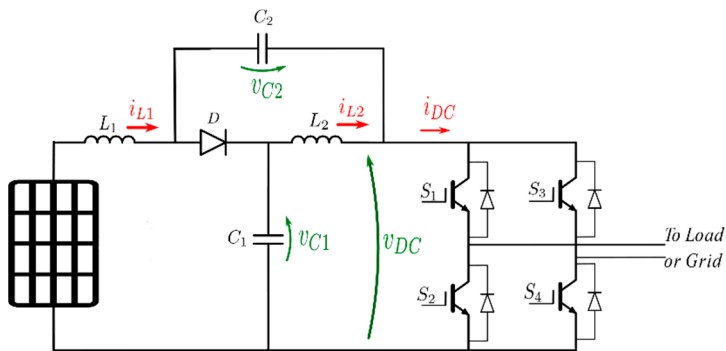

**Figure 4.** A single-phase qZSI.

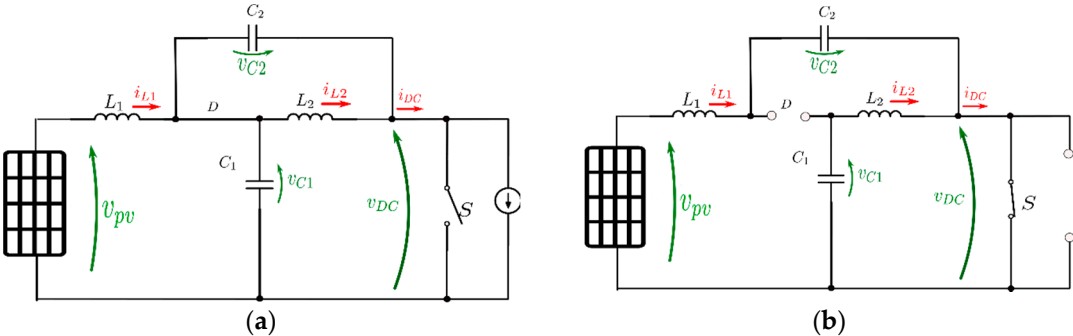

**Figure 5.** Equivalent circuits of the qZSI (**a**) in a non-shoot-through state and a (**b**) shoot-through state.

The input voltage and the inductors charge the capacitors and supply the load. During the ST state, as shown Figure 5b, the DC link of the inverter ($v_{dc}$) is short-circuited through the lower and upper switches of the qZSI; therefore, no energy is exchanged from the DC link to the load, and the capacitors charge the inductors.

For the system, the state vector $X$ and input vector $U$ are expressed as:

$$X = \begin{bmatrix} i_{L1} & i_{L2} & v_{C1} & v_{C2} \end{bmatrix}^T = \begin{bmatrix} x_1 & x_2 & x_3 & x_4 \end{bmatrix}^T \tag{4}$$

$$U = \begin{bmatrix} v_{pv} & i_{DC} \end{bmatrix}^T \tag{5}$$

where $i_{L1}$, $i_{L2}$, $v_{C1}$, and $v_{C2}$ are the inductor's currents and capacitor's voltages in the Z-source network; $i_{DC}$ is the DC-link current. Assume that $C = C_1 = C_2$ and $L = L_1 = L_2$. Define the ST interval $T_{st}$, NST interval $T_{nst}$, and switching period $T_s = T_{st} + T_{nst}$; thus, the ST–duty ratio is $d_{st} = T_{st}/T_s$, $d_{nst} = 1 - d_{st}$, and the switching frequency is $f_s = 1/T_s$. The state-space form, $\dot{X} = A_{st}X + B_{st}U$, of the differential equations in the ST mode is written as:

$$\frac{d}{dt} \begin{bmatrix} x_1 \\ x_2 \\ x_3 \\ x_4 \end{bmatrix} = \begin{bmatrix} 0 & 0 & 0 & \frac{1}{L} \\ 0 & 0 & \frac{1}{L} & 0 \\ 0 & -\frac{1}{C} & 0 & 0 \\ -\frac{1}{C} & 0 & 0 & 0 \end{bmatrix} \begin{bmatrix} i_{L1} \\ i_{L2} \\ v_{C1} \\ v_{C2} \end{bmatrix} + \begin{bmatrix} \frac{1}{L} & 0 \\ 0 & 0 \\ 0 & 0 \\ 0 & 0 \end{bmatrix} \begin{bmatrix} v_{pv} \\ i_{DC} \end{bmatrix} \tag{6}$$

Similarly, the state space form, $\dot{X} = A_{nst}X + B_{nst}U$, of the differential equations are written as:

$$\frac{d}{dt} \begin{bmatrix} x_1 \\ x_2 \\ x_3 \\ x_4 \end{bmatrix} = \begin{bmatrix} 0 & 0 & 0 & -\frac{1}{L} \\ 0 & 0 & -\frac{1}{L} & 0 \\ 0 & \frac{1}{C} & 0 & 0 \\ \frac{1}{C} & 0 & 0 & 0 \end{bmatrix} \begin{bmatrix} i_{L1} \\ i_{L2} \\ v_{C1} \\ v_{C2} \end{bmatrix} + \begin{bmatrix} \frac{1}{L} & 0 \\ 0 & 0 \\ 0 & \frac{-1}{C} \\ 0 & \frac{-1}{C} \end{bmatrix} \begin{bmatrix} v_{pv} \\ i_{DC} \end{bmatrix} \tag{7}$$

The average state-space dynamic model of the qZSI network was found as:

$$\dot{X} = AX + Bu(t) \tag{8}$$

$$A = A_{st}d_{st} + A_{nst}d_{nst} \tag{9}$$

$$B = B_{st}d_{st} + B_{nst}d_{nst} \tag{10}$$

$$A = \begin{bmatrix} \frac{1}{L_1}(v_{pv} - v_{C1}) \\ \frac{-1}{L_2}(v_{C2}) \\ \frac{1}{C_1}(i_{L1} - i_{DC}) \\ \frac{1}{C_2}(i_{L2} - i_{DC}) \end{bmatrix} \quad B = \begin{bmatrix} \frac{1}{L_1}(v_{C2} + v_{C1}) \\ \frac{1}{L_2}(v_{C2} + v_{C1}) \\ \frac{1}{C_1}(i_{DC} - i_{L1} - i_{L2}) \\ \frac{1}{C_2}(i_{DC} - i_{L1} - i_{L2}) \end{bmatrix} \tag{11}$$

In Equation (8), the $u(t)$ is the control input.

## 4. Sliding Mode Controller for the MPPT Unit

An SMC is well known as a nonlinear control method that is robust against parameter uncertainties and parameter variations in the system. In much of the literature, SMCs were applied to power electronic converters and power systems such as AC inverters, buck, and boost [27–30]. The SMC consists of two phases: in the first phase, an equilibrium surface is defined, and in the second phase, a discontinuous control law is designed.

### 4.1. Sliding Surface

In this paper, the sliding surface was defined based on the power curve of the PV. The MPP was obtained by solving Equation (12):

$$\frac{\partial P_{pv}}{\partial i_{pv}} = \frac{\partial \left( R_{pv} i_{pv}^2 \right)}{\partial i_{pv}} = i_{pv} \left( 2R_{pv} + i_{pv} \frac{\partial R_{pv}}{\partial i_{pv}} \right) \tag{12}$$

where $R_{pv} = v_{pv}/i_{pv}$. To satisfy Equation (12) and force the PV module to work at the MPP, the sliding surface was defined as in (13):

$$S = 2R_{pv} + i_{pv} \frac{\partial R_{pv}}{\partial i_{pv}} \tag{13}$$

Since $i_{pv}$ is equal to $i_{L1}$, it can be written according to Equation (14):

$$S = 2\frac{v_{pv}}{x_1} + x_1 \frac{\partial \frac{v_{pv}}{x_1}}{\partial x_1} \tag{14}$$

### 4.2. Equivalent Control

By defining the sliding surface, a control law should be obtained for the qZSI that enforces the system to move on the sliding surface. The following structure for the control input was used [33]:

$$u(t) = u_{eq}(t) + u_n(t) \tag{15}$$

where $u_{eq}(t)$ and $u_n(t)$ are known as the equivalent control input and nonlinear switching input, respectively. $u_{eq}(t)$ describes the behavior of the system on the sliding surface, and $u_n(t)$ moves the state of the system toward the sliding surface and keeps the state on the sliding surface in the presence of uncertainty. According to [16], the equivalent control is obtained as:

$$\dot{S} = \left[ \frac{\partial S}{\partial X} \right]^T \dot{X} = 0 \ \leftrightarrow \ u(t) = u_{eq}(t) \tag{16}$$

where

$$\left[ \frac{\partial S}{\partial X} \right]^T = \left[ \begin{array}{cccc} \frac{\partial S}{\partial x_1} & \frac{\partial S}{\partial x_2} & \frac{\partial S}{\partial x_3} & \frac{\partial S}{\partial x_4} \end{array} \right]^T \tag{17}$$

From Equations (14), (16) and (17), the following can be written:

$$\frac{\partial S}{\partial x_2} = \frac{\partial S}{\partial x_3} = \frac{\partial S}{\partial x_4} = 0 \tag{18}$$

$$\dot{S} = \left[ \frac{\partial S}{\partial X} \right]^T \dot{X} = \frac{\partial S}{\partial x_1} \dot{x}_1 \tag{19}$$

From Equations (8), (11) and (19), the equivalent control was calculated as:

$$\dot{S} = \frac{\partial S}{\partial x_1} \left( \frac{1}{L_1} (v_{pv} - x_3) + \frac{1}{L_1} (x_4 + x_3) u_{eq} \right) = 0 \tag{20}$$

The equivalent control was then derived by:

$$u_{eq} = \frac{x_3 - v_{pv}}{x_4 + x_3} \tag{21}$$

The value of $u_{eq}$ normally varies between 0 and 1, but in a qZSI converter, the shoot-through duty ratio should be limited to 0 to 0.3. Then, $u_n(t)$ is chosen so that the Lyapunov stability criteria ($\lim_{S \to 0} S\dot{S} < 0$) are met. The chosen $u_n(t)$ was as:

$$u_n(t) = \frac{v_{pv} - Mx_3}{x_4 + x_3} \tag{22}$$

where $M$ is the control signal, which is calculated through the Lyapunov stability criteria. Therefore, Equations (21) and (22) give the control law defined in Equation (15) as:

$$u(t) = \frac{1 - M}{x_4 + x_3} x_3 \tag{23}$$

*4.3. Stability Analysis*

The Lyapunov theory is known as the most effective approach for investigating the stability of a control system. According to this opinion, after the reaching phase, the following conditions must hold so that the state trajectories stay on the sliding manifold and slide along this manifold into the origin. A Lyapunov function and its time derivative are defined as [19]:

$$V = \frac{1}{2}S^2, \ dV = S\dot{S} \tag{24}$$

The following condition must be satisfied to warrant the existence of the SM operation:

$$dV = S\dot{S} < 0 \tag{25}$$

The time derivative of the Lyapunov function can be obtained as:

$$dV = \frac{\partial S}{\partial x_1}\left(\frac{1}{L_1}\left(v_{pv} - x_3\right) + \frac{1}{L_1}(x_4 + x_3)u(t)\right)S \tag{26}$$

By substituting Equation (23) into Equation (26), it can be calculated as:

$$dV = \frac{1}{L_1}\frac{\partial S}{\partial x_1}\left(v_{pv} - Mx_3\right)S \tag{27}$$

From Equations (13) and (14), it can be rewritten as:

$$\frac{\partial S}{\partial x_1} = 3\frac{\partial R_{pv}}{\partial x_1} + x_1\frac{\partial^2 R_{pv}}{\partial x_1{}^2} \tag{28}$$

To calculate the first derivative of the $R_{pv}$ to $x_1$, it can be written as:

$$\frac{\partial R_{pv}}{\partial x_1} = \frac{\partial\left(\frac{v_{pv}}{x_1}\right)}{\partial x_1} = \frac{1}{x_1}\frac{\partial v_{pv}}{\partial x_1} - \frac{v_{pv}}{x_1^2} \tag{29}$$

To calculate the second derivative of the $R_{pv}$ to $x_1$ and from Equation (29), it can be written:

$$\frac{\partial^2 R_{pv}}{\partial x_1{}^2} = \frac{1}{x_1}\frac{\partial^2 v_{pv}}{\partial x_1{}^2} + \frac{2v_{pv}}{x_1^3} - \frac{2}{x_1^2}\frac{\partial v_{pv}}{\partial x_1} \tag{30}$$

By placing Equations (29) and (30) in Equation (28) and rewriting it, the following equation is obtained:

$$\frac{\partial S}{\partial x_1} = \frac{1}{x_1}\frac{\partial v_{pv}}{\partial x_1} + \frac{\partial^2 v_{pv}}{\partial x_1{}^2} - \frac{v_{pv}}{x_1^2} \tag{31}$$

Using Equation (1) and considering that $i_{pv} = x_1$, the $v_{pv}$ and its first and second derivatives can be obtained as:

$$v_{pv} = \frac{k_b TA}{q} ln\left(\frac{I_{ph} + I_d - x_1}{I_d}\right) \tag{32}$$

$$\frac{\partial v_{pv}}{\partial x_1} = -\frac{k_b TA}{q}\frac{I_d}{I_{ph} + I_d - x_1}, \quad \frac{\partial^2 v_{pv}}{\partial x_1{}^2} = \frac{k_b TA}{q}\frac{I_d}{\left(I_{ph} + I_d - x_1\right)^2} \tag{33}$$

From Equations (31)–(33), it can be found that Equation (31) always has a negative sign, and for simplicity Equation (27) can be rewritten as:

$$dV = \gamma\left(v_{pv} - Mx_3\right)S < 0 \tag{34}$$

where

$$\gamma = \frac{1}{L_1}\frac{\partial S}{\partial x_1} < 0. \tag{35}$$

By substituting Equation (29) into Equation (14), the sliding surface can be calculated as:

$$S = \frac{v_{pv}}{x_1} + \frac{\partial v_{pv}}{\partial x_1} \tag{36}$$

Based on Equations (24)–(34), the control law M can be chosen as:

$$M = -\frac{x_1}{x_3}\frac{\partial v_{pv}}{\partial x_1} \tag{37}$$

To check the stability of the control function and by considering Equation (25) and Equations (34)–(37), the system mode can be divided into the following sections:

Case I: $S > 0,\ \gamma < 0 \rightarrow v_{pv} - Mx_3 > 0$

$$S > 0 \rightarrow \frac{v_{pv}}{x_1} + \frac{\partial v_{pv}}{\partial x_1} > 0 \rightarrow v_{pv} + x_1\frac{\partial v_{pv}}{\partial x_1} > 0 \rightarrow v_{pv} - \left(-\frac{x_1}{x_3}\frac{\partial v_{pv}}{\partial x_1}\right)x_3 > 0 \rightarrow v_{pv} - Mx_3 > 0 \tag{38}$$

Case II: $S < 0,\ \gamma < 0 \rightarrow v_{pv} - Mx_3 < 0$

$$S < 0 \rightarrow \frac{v_{pv}}{x_1} + \frac{\partial v_{pv}}{\partial x_1} < 0 \rightarrow v_{pv} + x_1\frac{\partial v_{pv}}{\partial x_1} < 0 \rightarrow v_{pv} - \left(-\frac{x_1}{x_3}\frac{\partial v_{pv}}{\partial x_1}\right)x_3 < 0 \rightarrow v_{pv} - Mx_3 < 0 \tag{39}$$

Considering Equations (38) and (39), it can be assured that the system has been stabilized according to Lyapunov's criteria.

## 5. Simulation and Experimental Results

To verify the effectiveness of the proposed control method a simulation model and an experimental prototype of a single-phase qZSI were constructed in MATLAB/ Simulink and in the laboratory. The overall block diagram of the system is illustrated in Figure 6, and the electrical components of the qZSI are given in Table 1. The parameters of the PV array are listed in Table 2. A prototype of the qZSI was developed in the laboratory using the Digital Signal Processor TMS320F28379D, as shown in Figure 7, and the system specifications are listed in Tables 1 and 2. To validate the SMC-PCPV and observe their different responses under different operational conditions, the SMC-PCPV was investigated in different environmental conditions. Hence, this section is divided into four subsections. The first subsection investigates the response of the SMC-PCPV in the simulation model

under a stepping change temperature. The performance of the proposed SMC under the stepping change of radiation in a simulation environment was studied in the second subsection. In the third subsection, the experimental results of the laboratory prototype in real conditions are presented. The dynamic response study of the SMC-PCPV and the existing controllers are presented in the fourth subsection. In this study, the modulation index was fixed at 0.7.

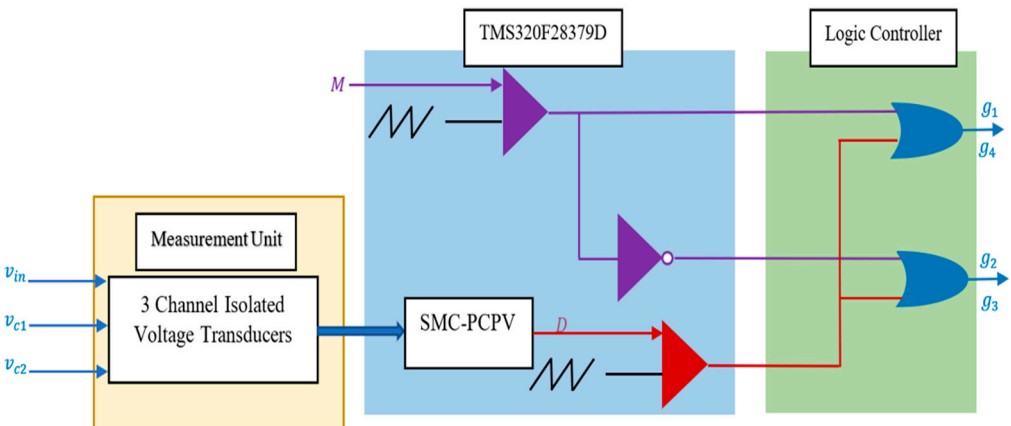

**Figure 6.** Overall block diagram of the control system.

**Table 1.** Parameters of the qZSI.

| Parameter | Value |
|---|---|
| qZSI inductors ($L_1, L_2$) | 500 μH |
| qZSI capacitors ($C_1, C_2$) | 3300 μF |
| Load resistance | 2.5 Ω |
| Load inductance | 200 μH |
| Modulation index | 0.7 |
| Switching frequency | 10 kHz |
| Output frequency | 50 H z |

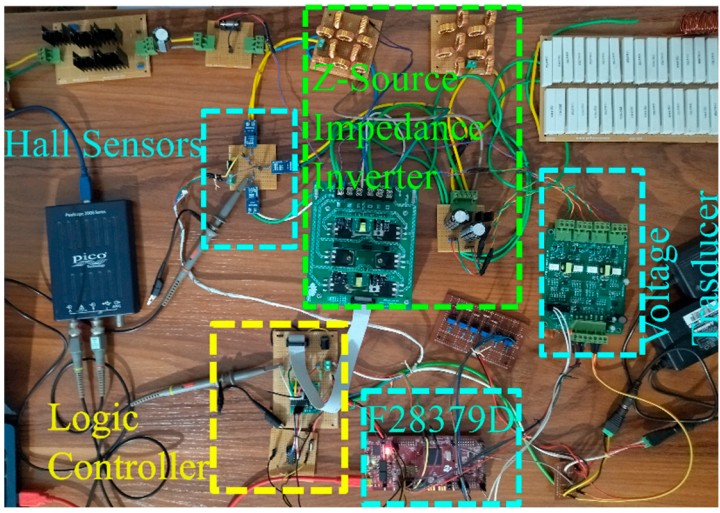

**Figure 7.** Prototype of the single-phase qZSI.

**Table 2.** Parameters of the PV array at T = 25 °C and G = 1 Sun.

| Parameter | Value |
|---|---|
| The voltage at the maximum power point, $V_{MPP}$ | 18.4 V |
| Current at the maximum power point, $I_{MPP}$ | 4.62 A |
| Open-circuit voltage, $V_{oc}$ | 21.9 V |
| Short-circuit current, $I_{sc}$ | 4.97 A |
| Maximum power | 85 W |

### 5.1. Response of the SMC-PCPV under Varying Temperatures

In this subsection, the response of the SMC-PCPV was investigated by changing the temperature. The temperature was first fixed at 25 °C and changed to 35 °C at 0.15 s. In this case, the irradiance level was held constant at 1000 W/m$^2$. The $v_{pv}$, $i_{pv}$, $P_{pv}$, $v_{c1}$, $v_{c2}$, and $i_{load}$ under changing temperatures are shown in Figures 8 and 9. It can be seen that in the case of exposure of the controller to a disturbance of temperature change type, the SMC-PCPV had good stability and could obtain the maximum power from the PV cells with high speed and accuracy. In this case, the rise time was 0.8 ms, and the settling time was 3.54 ms.

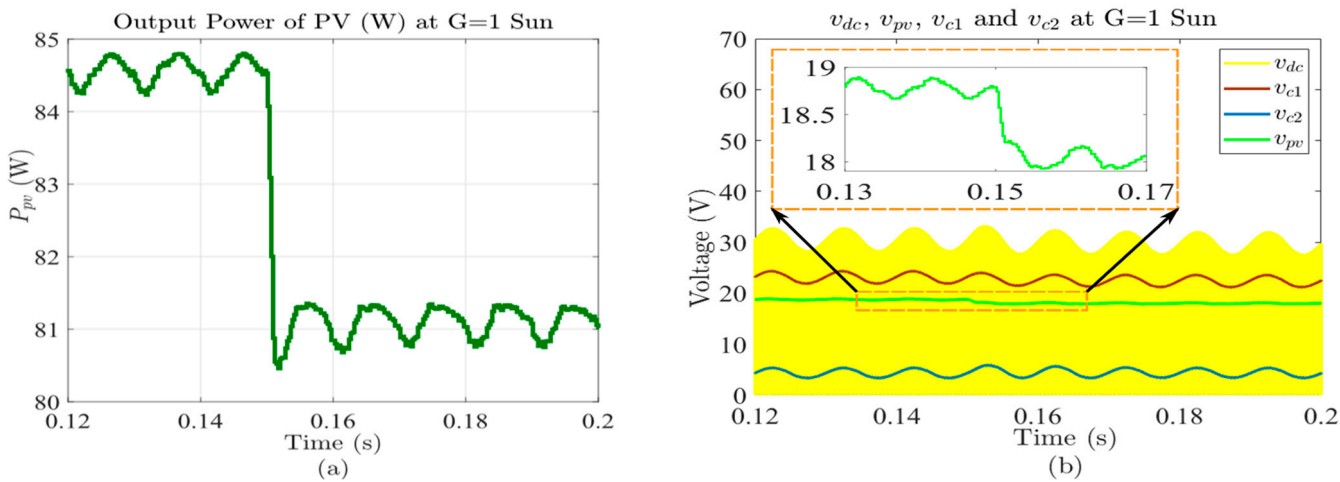

**Figure 8.** The waveforms of (**a**) $P_{pv}$ and (**b**) $v_{pv}$, $v_{dc}$, $v_{c1}$, and $v_{c2}$ under changing temperatures.

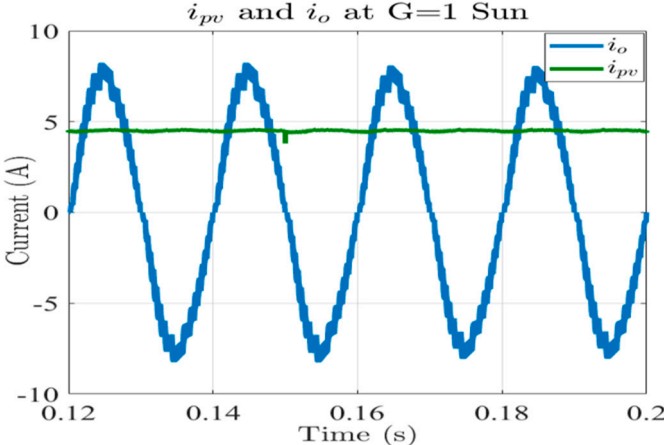

**Figure 9.** The simulation results of $i_{pv}$ and $i_{load}$ under changing temperatures.

The MPPs of the panel obtained from the simulation of the SMC-PCPV were 18.6 V, 4.5 A, and 83.7 W and 17.9 V, 4.5 A, and 80.55 W at T = 25 °C and T = 35 °C, respectively, when G was fixed at 1 Sun. The MPPs of the panel based on the manufacturer's information

were 18.4 V, 4.62 A, and 85 W and 17.63 V, 4.63 A, and 81.6 W at T = 25 °C and T = 35 °C, respectively, when $G$ = 1 Sun. The SMC-PCPV could absorb the maximum achievable power from the PV panel when the temperature changed.

### 5.2. Response of the SMC-PCPV under Changing Irradiance

In this subsection, the response of the SMC-PCPV was investigated by changing the irradiance. The irradiance level was first fixed at 1000 W/m$^2$, and it was changed to 850 W/m$^2$ at 0.2 s and then to 700 W/m$^2$ at 0.25 s. The temperature was held at 25 °C throughout the simulation. The $v_{pv}$, $i_{pv}$, $P_{pv}$, $v_{c1}$, $v_{c2}$, and $i_{load}$ are shown in Figures 10 and 11. It can be seen that by changing the radiation, the proposed controller can track the maximum achievable power. In other words, the introduced controller has the appropriate stability to deal with the perturbation of radiation change. The rise time of the proposed SMC was 0.9 ms.

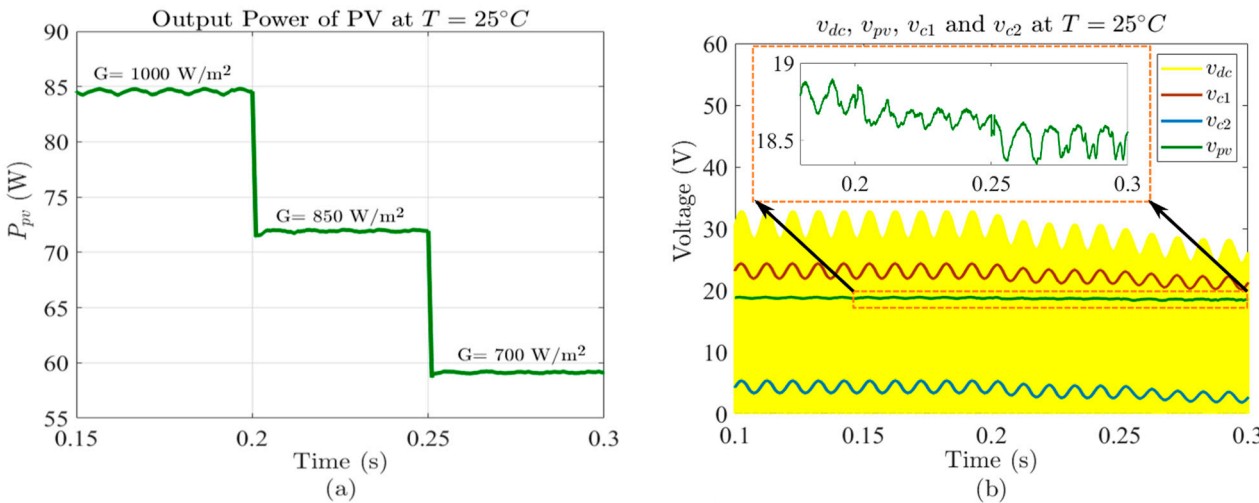

**Figure 10.** The simulation results of (**a**) $P_{pv}$ and (**b**) $v_{pv}$, $v_{dc}$, $v_{c1}$, and $v_{c2}$ under changing irradiance.

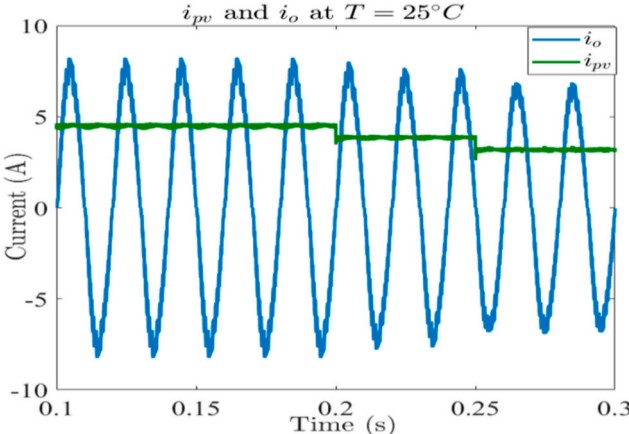

**Figure 11.** The simulation results of $i_{pv}$ and $i_{load}$ under changing irradiance.

The MPPs of the panel obtained from the simulation of the SMC-PCPV were 18.6 V, 4.5 A, and 83.7 W; 18.4 V, 3.85 A, and 70.8 W; 18.3 V, 3.15 A, and 57.62 W at 1000 W/m$^2$, 850 W/m$^2$, and 700 W/m$^2$, respectively, when the temperature was fixed at 25 °C. The MPPs of the panel based on the manufacturer's information were 18.4 V, 4.62 A, and 85 W; 18.3 V, 3.93 A, and 72 W; 18.3 V, 3.24 A, and 59.2 W, respectively, at the mentioned conditions. The simulation results of the introduced controller are shown in Table 3. The maximum difference between the absorbed power ($P_{absd}$) and the absorbable power ($P_{able}$) under different simulated conditions was equal to 2.65%. The simulation results show

that the controller introduced in this paper can track changes in irradiance intensity and temperature.

**Table 3.** Simulation results of the SMC-PCPV.

| Irradiance Level | 1000 W/m$^2$ | | 850 W/m$^2$ | 700 W/m$^2$ |
|---|---|---|---|---|
| Temperature | **25 °C** | **35 °C** | **25 °C** | |
| $V_{MPP}$ | 18.6 V | 17.9 V | 18.4 V | 18.3 V |
| $I_{MPP}$ | 4.5 A | 4.5 A | 3.85 A | 3.15 A |
| $P_{absd}$ | 83.7 W | 80.55 W | 70.84 W | 57.62 W |
| $P_{able}$ | 85 W | 81.6 W | 72 W | 59.2 W |
| $\frac{(P_{able}-P_{absd})}{P_{able}} \times 100$ | 1.52% | 1.35% | 1.6% | 2.65% |

*5.3. Experimental Results of the SMC-PCPV*

The prototype of the qZSI with an SMC was built based on the previous statements. The parameters of the prototype were the same as the parameters applied in the simulations. The control signals were generated from the digital signal processor (DSP) TMS320F28379D from Texas Instruments. A digital oscilloscope was used to observe the experimental results, and to better display the results, the obtained data were imported into MATLAB software and then presented. To test the introduced controller, a TPS105S-85W solar panel was tested in real conditions. The values of $V_{oc}$ and $I_{sc}$ of the panel were measured at different radiations and are shown in Table 4. According to the characteristic curve of the panel and the measured values of $V_{oc}$ and $I_{sc}$, the $P_{able}$, $V_{MPP}$, and $I_{MPP}$ were estimated [34] and are presented in Table 4. The experimental results for each case are presented in Table 4. In the second column of Table 4, it is specified that the mentioned quantity was obtained through measurement (Me) or modeling and estimation (Es). In the case of A, the $V_{oc}$ and $I_{sc}$ were equal to 21.6 V and 3.7 A, respectively. According to the modeling condcuted on the panel and its characteristic curve, the $P_{able}$ can be estimated at 62.5 W. The amount of $P_{absd}$ from the prototype, in this case, was equal to 60 W. The difference between the $P_{able}$ and the $P_{absd}$, in this case, was equal to 2.5 W, or in other words, the absorbed power had a deviation of 4% from the maximum absorbable power, where in the case of B, the deviation between the $P_{able}$ and the $P_{absd}$ reached 3%. The experimental results obtained from the prototype showed that the introduced controller cold control the qZSI in such a way that it absorbed the maximum power from the PV cell with an acceptable error. The waveforms of $v_{dc}$, $v_{pv}$, $i_{pv}$, $v_{c1}$, $v_{c2}$, and $i_{load}$ associated with case A that were obtained from the experimental test are shown in Figure 12.

**Table 4.** Parameters of PV array at environmental conditions.

| | $V_{oc}$ | $I_{sc}$ | $V_{MPP}$ | $I_{MPP}$ | $P_{able}$ | $V_{MPP}$ | $I_{MPP}$ | $P_{absd}$ | $\frac{(P_{able}-P_{absd})}{P_{able}} \times 100$ |
|---|---|---|---|---|---|---|---|---|---|
| Estimated/measured | Me [1] | Me | Es [2] | Es | Es | Me | Me | Me | |
| Case A | 21.6 V | 3.7 A | 18.3 V | 3.42 A | 62.5 W | 18.2 V | 3.2 A | 60 W | 4% |
| Case B | 21.3 V | 2.75 A | 18.19 V | 2.54 A | 46.2 W | 17.92 V | 2.5 A | 44.8 W | 3% |

[1] Measured value. [2] Estimated value.

In this paper, the aim was to show the efficiency of the introduced controller in absorbing the maximum absorbable power of the panel in different weather conditions, where the absorbed power had a deviation of 4% and 1.5% from the maximum absorbable power in the experimental and simulation results, respectively. The difference between the maximum power absorbed in the simulated and experimental modes was related to the losses of the 50 m interface cable between the solar panel and the converter in the experimental mode. Most modern MPPTs are approximately 93–98% efficient in the conversion [35,36].

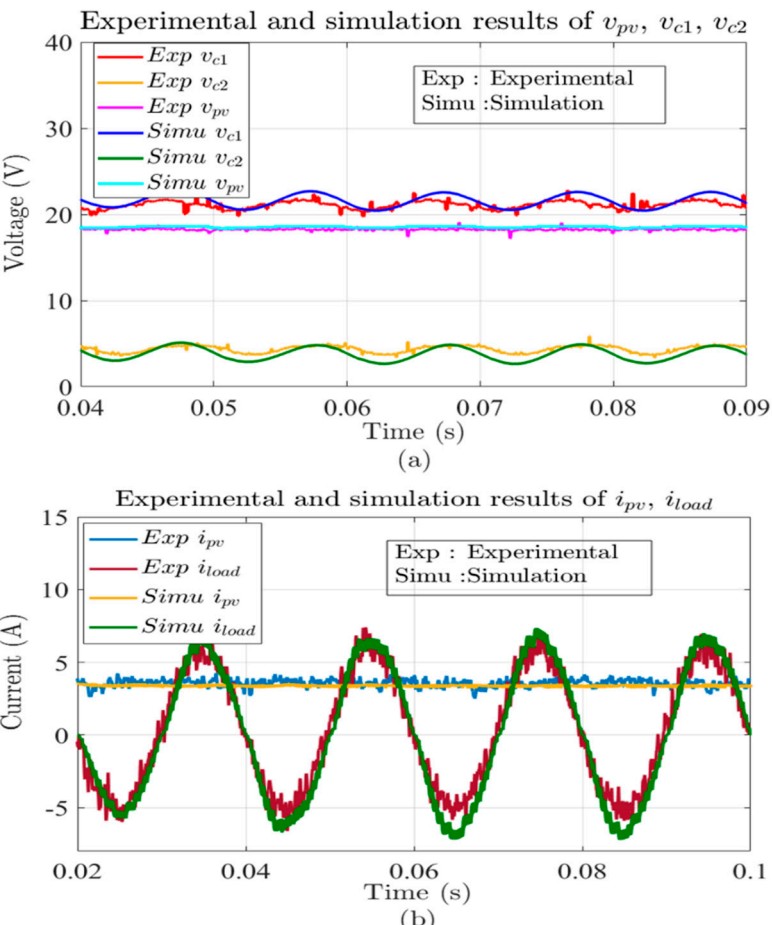

**Figure 12.** The experimental and simulation results (**a**) $v_{pv}$, $v_{dc}$, $v_{c1}$, and $v_{c2}$ and (**b**) $i_{pv}$ and $i_{load}$ associated with case A.

### 5.4. Comparison of the Dynamic Response

In this part, a comparison between the dynamic response of the SMC-PCPV and the nonlinear MPPT controllers [1] is made. In [1], the dynamic characteristics of four controllers are given, which were the fuzzy logic-based nonlinear controller (FLBC), PSO, and integral back-stepping (IBS). The dynamic values of the nonlinear MPPT controllers [1] and the proposed controller are given in Table 5, where the SMC-PCPV was evaluated under the standard temperature of 25 °C and radiation of 1000 W/m². The table was formulated by comparing the analyzed techniques on the basis of rise time (RT), settlingtime (ST) (2% and 5% criteria), and ripples in $v_{pv}$. According to Table 5, it can be seen that the proposed controller had the fastest RT among other nonlinear controllers. It also had an acceptable ST and ripple.

**Table 5.** Comparison of the dynamic responses.

| Method | RT (ms) | ST 5% Criteria (ms) | ST 2% Criteria (ms) | $\frac{\Delta v_{pv}}{V_{pv}}*100$ |
|---|---|---|---|---|
| Backstepping [37] | 2.42 | 3.1 | 3.8 | 0.25% |
| PSO [38] | 2.22 | 19 | NA | 6.5% |
| IBS [1] | 2.17 | 2.9 | 3.2 | 0.23% |
| FLBC [39] | 2.17 | 3 | 8.4 | 1.2% |
| SMC-PCPV | 0.9 | 3.1 | 3.84 | 1.07% |

## 6. Conclusions

Obtaining the maximum power from a PV panel under changing environmental conditions is an ongoing challenge. In this paper, an SMC based on the PV power curve

(SMC-PCPV) was presented to receive the maximum power from the solar panel. The stability of the system was ensured using the Lyapunov stability criterion. The SMC-PCPV was implemented on a qZSI. The qZSI is a power conditioner that employs a Z-source network for boosting the photovoltaic voltage and connecting to the inverter as a single stage. By controlling the ST duty cycle, the objective of tracking MPP can be carried out. To verify the proposed controller, the SMC-PCPV was carried out on computer simulation and laboratory prototype. The SMC-PCPV was investigated at different environmental conditions such as varying temperature and irradiance. The simulation and experimental results showed that the SMC-PCPV had an acceptable overall performance. The most prominent feature of this controller was its high speed in response to changes in input parameters. Other features of this proposed controller include ease of implementation, stability against disturbances and environmental changes, and high efficiency. Compared to other nonlinear controllers presented in the articles, this controller had a rise time equal to approximately half of the other controllers. The maximum error rate of the received power with the maximum achievable power was less than 4%, which is an acceptable value. It should be noted that the power obtained at the input of the converter was the basis of the calculation, which eliminated the loss of 50 m of cable between the panel and the converter, and the controller efficiency increased to 97.4%, which is one of the methods with high efficiency. Due to the inductor current ripple of the qZSI not being optimal in most applications and the fact that large inductor current ripples will result in higher inductor band switching losses, which further leads to distortions in the output current, the other prominent feature of this controller is the almost constant current that is drawn from the PV panel. In the future, this work can be extended by double frequency voltage ripple suppression.

**Author Contributions:** Conceptualization, J.M.A.; methodology, J.M.A.; software, J.M.A.; validation, J.M.A., T.N.; formal analysis, J.M.A.; investigation, J.M.A.; resources, J.M.A.; data curation, J.M.A.; writing—original draft preparation, J.M.A., T.N.; writing—review and editing, J.M.A., T.N., M.I.; visualization, J.M.A., T.N.; supervision, T.N.; project administration, M.I. All authors have read and agreed to the published version of the manuscript.

**Funding:** This research received no external funding.

**Data Availability Statement:** The study did not report any data.

**Conflicts of Interest:** The authors declare no conflict of interest.

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
