# Peer review of "Applying a Sliding Mode Controller to Maximum Power Point Tracking in a Quasi Z-Source Inverter Based on the Power Curve of a Photovoltaic Cell"

_electronics, doi:10.3390/electronics11142164_

Round 1
Reviewer 1 Report
The authors studied “Applying Sliding Mode Controller to Maximum Power Point Tracking in Quasi Z-source Inverter Based on the Power Curve of Photovoltaic Cell”. The work is interesting and presented with a lot of supporting evidence. In my opinion lot of typographical errors. The author must be addressed the following comments and need careful revision of the entire manuscript.
The structure of the abstract and introduction chapter is not standard.
Minor spelling mistakes, it is difficult to interpret. The purpose is not entirely clear, the introduction does not reveal what the authors want to prove.
The measured and simulated results should be in common figure for better approximation. The axis labels are difficult to read in many figures.
Why is the high demand for data management a problem, if the process presented by the authors uses less data, then not more inaccurate?
The qZSI has not been compared with other currently applicable solutions / converters. This solution could have been compared with LLC resonant converters. We suggest an article as a reference in this topic: https://doi.org/10.3390/en15030708
What is the maximum solar power that these qZSI converters can handle? The efficiency of the tested converter was not shown.
In the presentation of the other MPPT solutions, did you describe how the values oscillate around the theoretical working point, to what extent the oscillation of the presented solution is compared to the other solutions?
What was the measurement error, what was the class accuracy of the instruments used?
The switching frequency is 10 kHz, why? In practice a higher frequency is used.
In the measurement setup (Figure 7), the ground wire of the passive probes of the oscilloscope is not connected. What is the reason?
Why 85 W monocrystalline solar panels have been studied, larger and polycrystalline panels are typically used today. Could you the values obtained transfer to higher-performance panels?
Author Response
Response to Reviewer 1 Comments for paper Ref. electronics-1748389
Point 1: The structure of the abstract and introduction chapter is not standard.
Response 1: Thank you for the comment. The structure of the abstract and introduction was revised.
Point 2: Minor spelling mistakes, it is difficult to interpret. The purpose is not entirely clear, the introduction does not reveal what the authors want to prove.
Response 2: Thank you for mentioning. By revising the introduction section, an attempt has been made to clearly state the purpose of the article.
Point 3: The measured and simulated results should be in common figure for better approximation. The axis labels are difficult to read in many figures.
Response 3: Thanks for your careful look. In Figures 8.b and 10.b , the axes were not legible and clear, which was corrected. If you have a specific shape, we would appreciate it if you could announce it, so that it can be corrected. The measured and simulated results are shown in one figure for better comparison.
Point 4: Why is the high demand for data management a problem, if the process presented by the authors uses less data, then not more inaccurate?
Response 4: Thank you for mentioning. Unlike intelligent methods, which require large amounts of information to identify the system and overcome disturbances, and changing system parameters requires re-identifying the system. The Sliding mode control (SMC) consists of an algorithm that is inherently resistant to changes in parameters, nonlinear models, external disturbances, and uncertainty . https://doi.org/10.1016/B978-0-12-818365-6.00005-7
Point 5: The qZSI has not been compared with other currently applicable solutions / converters. This solution could have been compared with LLC resonant converters. We suggest an article as a reference in this topic: https://doi.org/10.3390/en15030708.
Response 5: Thank you for your helpful comment. The requested change is applied and the relevant reference was added.
Point 6: What is the maximum solar power that these qZSI converters can handle? The efficiency of the tested converter was not shown.
Response 6: Thank you for mentioning. We used the FGA15N120 in qZSI, which is a high voltage IGBT with a collector voltage of 1200 V and a continuous collector current of 30A. The prototype is capable of handling solar panels with a power of more than one kilowatt. However, the maximum power tested with a solar panel is 200W. QZSI converters have low losses and high efficiency due to their single-stage structure, which has been shown in many articles. The efficiency of the system in this case is close to 92%. In this paper, the aim is to show the efficiency of the introduced controller in absorbing the maximum absorbable power of the panel in different weather conditions that the absorbed power has a deviation of 4% and 1.5% from the maximum absorbable power in experimetal and simulation results respectively. The difference between the maximum power absorbed in the simulated and experimental modes is related to the losses of the 50-meter interface cable between the solar panel and the converter in the experimental mode. Due to the fact that the solar simulator is being built by the team of the author of the article based on TPS105S-85W-MONO, this panel has been used to test the proposed controller to will be refrenced in future articles.
Point 7: In the presentation of the other MPPT solutions, did you describe how the values oscillate around the theoretical working point, to what extent the oscillation of the presented solution is compared to the other solutions?
Response 7: Thank you for the comment. The amount of power and voltage oscillation around the working point is clearly shown in Figure 6. In Table 5, the output voltage ripple of the PV panel around the operating point are added according to the various algorithms reported in reference [1].
Point 8: What was the measurement error, what was the class accuracy of the instruments used?
Response 8: Thank you for mentioning. The following equipment is used to measure voltage and current parameters.
|
Instrument |
Part Number |
accuracy |
AC/DC Digital Clamp Meters |
kyoritsu KT203 |
±3.0%rdg±8dgt (0 - 40.00A DC) ±1.5%rdg±5dgt (0-600V DC) |
|
Hall Effect-Based Linear Current Sensor |
ACS712/20A |
1.5% Total Output Error |
|
4 channel linear isolated voltage transducer based on A7840. |
LM115-02 |
1% Total Output Error |
|
Digital Multimeter |
HIOKI 3256-50 |
±0.6%rdg±2dgt (0-56V DC) |
|
Scope |
PicoScope 2204A |
|
Point 9: The switching frequency is 10 kHz, why? In practice a higher frequency is used.
Response 9: Thank you for your helpful comment. In most cases, the switching frequency in the qZSI is between 10 and 20 kHz (doi: 10.1109/ACCESS.2021.3073621). Although in a few articles, researchers used high-speed GaN switches to increase the switching frequency of the qSZI (doi: 10.1109/TPEL.2012.2219556). Moreover, the impedance network used in the qZSI, is the most voluminous part for which the size is determined by the ST interval. Increasing the switching frequency reduces the volume of the impedance network and consequently increases the efficiency of the converter.Thanks to the utilization of GaN FETs, the qZSI achieves a high efficiency at 100kHz switching frequency. The maximum reported efficiency of the converter is 98.06%. However, utilization of such expensive switches, increases the total price of the system, dramatically. For example, the cost of a GaN MOSFET TPH3205WSBQA-ND is 25.32$, whereas the cost for FCH104N60F is 4.6$. Therefore, there always exists a trade-off between the size, the efficiency and price of the inverter, mostly because of the ST state. According to what was mentioned, increasing the frequency has little effect on the operation of the SMC-PCPV, which is the main purpose of this article.
Point 10: In the measurement setup (Figure 7), the ground wire of the passive probes of the oscilloscope is not connected. What is the reason?
Response 10: Thank you for mentioning. By zooming in on Figure 7, you can see the a pin connected to the ground connection of Probe A. This pin is connected to the ground of circuit. By connecting the ground of probe A to this pin, the ground of probe B is internally connected to the ground point and there is no need to connect to the ground point by probe b.
Point 11: Why 85 W monocrystalline solar panels have been studied, larger and polycrystalline panels are typically used today. Could you the values obtained transfer to higher-performance panels?
Response 9: Thank you for your helpful comment. Due to the widespread use of monocrystalline panels in our region due to the low price in real applications and in the panel in question is one of the most widely used in low power applications. The team of authors of the article are building a simulator of this panel, in order to be able to compare what is happening in real conditions.The reason for using this panel in this article is for comparison and reference in future articles. Higher power panels and more up-to-date technologies with better performance, as the respected reviewer pointed out, can also be used in this structure and examined in the future.

Reviewer 2 Report
In this work, the authors proposed a sliding mode controller based on the power curve of photovoltaics. They then used this controller to find the maximum power point of the photovoltaics panel. Features of this controller were demonstrated using both simulation and experimental methods. While generally well-written, this manuscript has occasional typos, e.g., on page 1. “replacing renewable energy with traditional energy” should be “replacing traditional energy with renewable energy”. My main conservation of this manuscript is it is not sufficiently novel. To my best knowledge, no equation in this manuscript is new. A secondary point is that the authors attempted to prove that the controller has “a good speed to respond to the changing environmental condition”. By “environmental condition”, the authors referred to temperature and radiation. However, the authors did not actually vary the temperature in their experiments nor did they subject any equipment or devices to irradiation. As a result, I do not think this manuscript warrants publication in this journal.
Author Response
Thanks for the comment dear reviewer. It should be noted that the results of simulation and experimental prototype in the case of no change in voltage and temperature, have a very low error and values are very close to each other, which indicates the correctness of the simulation results. In the case of changing the radiation and temperature, the results of the simulation are presented, which indicates the efficiency of the converter in the face of changing environmental conditions. It can be concluded that the introduced controller is able to control the converter in the real case due to the simulation results and the similarity between the two results.
Reviewer 3 Report
The number of refrences of this paper entitled : Applying Sliding Mode Controller to Maximum Power Point Tracking in Quasi Z-source Inverter Based on the Power Curve of Photovoltaic Cell, is good, however, this reviewer wonders why there is not any citation published in this year 2022. The paper includes a cocktail of control theory and electronic materials. However, if one read the conclusions, all of them are biased to control systems, as mentioned in the sentences in section 6. Conclusion:
Obtaining maximum power from a PV panel under changing environmental condi-tions is an ongoing challenge. This is emphasized in the abstract in the first sentences when you mention: Abstract: Due to the nonlinear nature of Photovoltaics (PV) cells and the dependence of the maxi-mum achievable power on environmental conditions, a robust nonlinear controller is essential to warrant maximum power point tracking (MPPT) by managing the nonlinearities of the system and making it robust against varying environmental conditions. In all the manuscript you do not compare your work with references, you just compare with methods, as seen in Table 4. Comparison of dynamic response. Can you include the references associated to those methods and detail the advantages of your proposed one SMC-PCPV?
You mention something interesting in the following sentence in the abstract:
In this paper, a robust nonlinear Sliding Mode Controller based on the Power Curve of PV (SMC-PCPV) is proposed to find the Maximum Power Point (MPP) of a PV panel, for a Quasi Z-Source Inverter (qZSI) as a single-stage inverter. Can you include a summary, or a Table of Works realted to find the MPP with that Quasi Z-Source Inverter (qZSI) as a single-stage inverter?. Can your proposed SMC-PCPV be applied to other inverters and with more stages?
A tipo is given in the sentence:
To show the effectiveness and robustness of the proposed scheme, the SMC_PCPV has been carried out on computer simulations and laboratory prototypes. You can check that in all cases you write SMC-PCPV instead of SMC_PCPV.
You have three keywords:
Keywords: maximum power point tracking (MPPT); sliding mode controller (SMC); Quasi Z-Source Inverter (qZSI);… can you add Photovoltaics (PV) cells? And also can you discuss the nonlinear behavior, and if you use that nonlinear nature or do you suggest that linearizing is better to improve the control system?
When you talk about inverters in the sentence:
To obtain power from PV and inject it into the load various configurations have been pro-posed, which can be mentioned as one-stage and two-stage topologies [4, 5]. Single-stage inverters have received more attention than two-stage inverters due to their lower com-ponents and price, smaller size, simplicity, and higher efficiency [6–8]… Can you include a Table or a summary of main advantages that are approprite to introduce your method called SMC-PCPV?
In the thrid paragraph of the introduction you also mention some keys that may be summarized in a table: There are many different approaches to the MPPT algorithm ranging from simple to ad-vanced ones that have been reported in the literature in the ZSI inverter family [13, 14]. The MPPT methods can be divided into two classes; based on the gradient classic methods and based on intelligent methods… Can you discuss in your experiments and conclusions the advantage of your method and what are you improving with respect to the other clases of MPPT?
You are encouraged to include references in section 3. Mathematical Model of qZSI, you did not include any. The same is for the subsections 4.1. sliding surface, 4.2 and so on
At the end of page 9: Digital Signal Processor TMS320F28379D as shown in Figure 7, where the system specifics are listed in Table 1, and Table 2… you mean specifications? Instead of specifics
In your conclusions you need to clarify if 4% is good or not: The error rate of the received power with the maximum achievable power is less than 4%. Another prominent feature of this controller is the almost constant current that is drawn from the PV panel… Also, you may discuss if the invertir improves due to the almost constant current that is drawn from the PV panel
Author Response
Response to Reviewer 3 Comments for paper Ref. electronics-1748389
Point 1: The number of refrences of this paper entitled : Applying Sliding Mode Controller to Maximum Power Point Tracking in Quasi Z-source Inverter Based on the Power Curve of Photovoltaic Cell, is good, however, this reviewer wonders why there is not any citation published in this year 2022. The paper includes a cocktail of control theory and electronic materials. However, if one read the conclusions, all of them are biased to control systems, as mentioned in the sentences in section 6. Conclusion.
Response 1: Thank you for the comment. Regarding the comment of the dear reviewer about the absence of articles from 2022, it is worth mentioning that this article was written at the beginning of 2022 and was registered in the journal Electronics due to a delay. There have been several articles in the references section of this article since 2021. Following request from one of the reviewers to explain the one-stage, two-stage structure, and resonance converters, the reference has been selected from 2022. In the introduction and conclusion section, the requested changes of the dear reviewer were added.
Point 2: Obtaining maximum power from a PV panel under changing environmental condi-tions is an ongoing challenge. This is emphasized in the abstract in the first sentences when you mention: Abstract: Due to the nonlinear nature of Photovoltaics (PV) cells and the dependence of the maxi-mum achievable power on environmental conditions, a robust nonlinear controller is essential to warrant maximum power point tracking (MPPT) by managing the nonlinearities of the system and making it robust against varying environmental conditions. In all the manuscript you do not compare your work with references, you just compare with methods, as seen in Table 4. Comparison of dynamic response. Can you include the references associated to those methods and detail the advantages of your proposed one SMC-PCPV?
Response 2: Thank you for mentioning. The references are added to Table 5. It should be noted that in the text of our article, it is mentioned that the values in Table 5 are given from reference 1. The main advantages of SMC-PCPV are stability, strength against parameter changes, fast dynamic response, ease of implementation and no need to identify the system.
Point 3: You mention something interesting in the following sentence in the abstract:
In this paper, a robust nonlinear Sliding Mode Controller based on the Power Curve of PV (SMC-PCPV) is proposed to find the Maximum Power Point (MPP) of a PV panel, for a Quasi Z-Source Inverter (qZSI) as a single-stage inverter. Can you include a summary, or a Table of Works realted to find the MPP with that Quasi Z-Source Inverter (qZSI) as a single-stage inverter?. Can your proposed SMC-PCPV be applied to other inverters and with more stages?
Response 3: In the introduction section, paragraphs 4 and 5, the previous work done to find the MPP in qZSI is presented in detail. This method can also be applied to other converters and topologies.
Point 4: A tipo is given in the sentence:
To show the effectiveness and robustness of the proposed scheme, the SMC_PCPV has been carried out on computer simulations and laboratory prototypes. You can check that in all cases you write SMC-PCPV instead of SMC_PCPV.
Response 4: Thank you for your helpful comment. In all sections of the article, SMC_PCPV was replaced by SMC-PCPV.
Point 5: You have three keywords:
Keywords: maximum power point tracking (MPPT); sliding mode controller (SMC); Quasi Z-Source Inverter (qZSI);… can you add Photovoltaics (PV) cells? And also can you discuss the nonlinear behavior, and if you use that nonlinear nature or do you suggest that linearizing is better to improve the control system?
Response 5: Thank you for your comment. The very appropriate and correct suggestion of the respected reviewer in adding the word photovoltaic to the keywords pleases the authors of this article and has been added to the list of keywords. Section 2 describes the nonlinear nature of the solar panel. In this paper, linearization is not used to improve the controller, but according to the nonlinear equations of the solar panel, the Lyapunov criterion and SMC are used to find the MPP.
Point 6: When you talk about inverters in the sentence:
To obtain power from PV and inject it into the load various configurations have been pro-posed, which can be mentioned as one-stage and two-stage topologies [4, 5]. Single-stage inverters have received more attention than two-stage inverters due to their lower com-ponents and price, smaller size, simplicity, and higher efficiency [6–8]… Can you include a Table or a summary of main advantages that are approprite to introduce your method called SMC-PCPV?
Response 6: Thank you for mentioning. SMC-PCPV method is a robust controller to absorb maximum power from solar panels that can be applied to other converters, inverters and topologies. The most prominent feature of this controller are its simplicity of implementation and stability against system changes. Another main feature of this method is its high speed in response to changes in input parameters, which can be improved in the future by combining with intelligent methods.
Point 7: In the thrid paragraph of the introduction you also mention some keys that may be summarized in a table: There are many different approaches to the MPPT algorithm ranging from simple to ad-vanced ones that have been reported in the literature in the ZSI inverter family [13, 14]. The MPPT methods can be divided into two classes; based on the gradient classic methods and based on intelligent methods… Can you discuss in your experiments and conclusions the advantage of your method and what are you improving with respect to the other clases of MPPT?
Response 7: Thank you for your helpful comment. The most prominent feature of this controller is its high speed in response to changes in input parameters. Another features of this proposed controller include ease of implementation, stability against disturbances and environmental changes, and high efficiency. Compared to other nonlinear controllers presented in the articles, this controller has a rise time equal to about half of the other controllers.
Point 8: You are encouraged to include references in section 3. Mathematical Model of qZSI, you did not include any. The same is for the subsections 4.1. sliding surface, 4.2 and so on
Response 8: Thank you for your helpful comment. The effective opinion of the dear reviewer was applied in the article.
Point 9: At the end of page 9: Digital Signal Processor TMS320F28379D as shown in Figure 7, where the system specifics are listed in Table 1, and Table 2… you mean specifications? Instead of specifics
Response 9: Thanks for your careful look. The desired word correction was performed.
Point 10: In your conclusions you need to clarify if 4% is good or not: The error rate of the received power with the maximum achievable power is less than 4%. Another prominent feature of this controller is the almost constant current that is drawn from the PV panel… Also, you may discuss if the invertir improves due to the almost constant current that is drawn from the PV panel
Response 10: Thank you for your helpful comment. Descriptions of the converter efficiency and constant current that is drawn from the PV panel was added to the article.

Round 2
Reviewer 1 Report
I accept.